# Systemic immune-inflammatory indicators and bone mineral density in chronic kidney disease patients: A cross-sectional research from NHANES 2011 to 2018

**Yuying Jiang**, **Xiaorong Bao** *

Department of Nephrology, Jinshan Hospital, Fudan University, Shanghai, China

* xrbao19660108@163.com

**Data Availability Statement:** Publicly available datasets were analyzed for this study from the NHANES database (www.cdc.gov/nchs/nhanes/).

## Abstract

### Background

The purpose of this study was to look at the relationship between the Systemic Immune Inflammatory Index (SII) and bone mineral density (BMD) in the pelvis, left upper and lower limbs, lumbar spine, thoracic spine, and trunk in a chronic kidney disease (CKD) population in the United States.

### Methods

The National Health and Nutrition Examination Survey (2011–2016) yielded 2302 people with CKD aged >18 years. CKD was defined as eGFR less than 90 ml/min/1.73 $m^2$ or eGFR greater than 90 ml/min/1.73 $m^2$ with urine ACR greater than 30 mg/L.SII was calculated as PC * (NC / LC) from platelet count (PC), neutrophil count (NC), and lymphocyte count (LC). Multiple logistic regression was used to examine the relationship between BMD and SII at different sites in CKD patients, smoothed curve-fitting and generalized weighting models were used to investigate non-linear relationships, and a two-tailed linear regression model was used to find potential inflection points in the model.

### Results

We discovered a negative correlation between SII and pelvic BMD among 2302 participants after controlling for gender, age, and race [β = -0.008; 95% confidence value -0.008; 95% confidence interval (CI) -0.014, -0.002]. Lower PEBMD was related to increasing SII (trend p = 0.01125). After additional correction, only pelvic BMD remained adversely linked with SII [value -0.006; 95% CI -0.012, -0.000, p = 0.03368]. Smoothed curve fitting revealed a consistent inverse relationship between SII and pelvic BMD. Further stratified analyses revealed a substantial positive negative connection between SII and pelvic BMD in individuals who did not have hypertension, diabetes, a BMI of more than 30 kg/$m^2$, or stage 2 CKD. The connection between SII and PEBMD in people without diabetes revealed a strong inverted U-shaped curve.

**Funding:** The author(s) received no specific funding for this work.

**Competing interests:** The authors have declared that no competing interests exist.

## Conclusion

In individuals with CKD in the United States, there was a negative connection between the systemic immunoinflammatory index (SII) and pelvic BMD. The SII might be a low-cost and simple test for CKD-related BMD loss.

## Background

More than 10% of the world's population and more than 80 billion people are affected by chronic kidney disease (CKD), one of the world's leading non-communicable causes of death [1]. By 2024, it is expected to rank as the fifth leading cause of life expectancy loss globally [2]. Patients with CKD frequently develop mineral bone disorders, such as osteoporosis and renal osteodystrophy, which worsen with deteriorating renal function, which is extremely common and harmful [3]. The 2017 KDIGO recommendations state that when evaluating the diagnostic and treatment requirements for osteoporosis in individuals with CKD, testing BMD may be preferred to doing a bone biopsy [4]. Furthermore, a 2022 meta-analysis showed that lower BMD values were associated with an increased risk of all-cause mortality in patients with CKD [5]. In patients with CKD, the clinical utility of BMD measurements for assessing bone loss and other related health conditions remains worth investigating due to the presence of various hormonal and metabolic alterations [6].

In fact, ongoing low-grade inflammation and immunological dysfunction are now recognized as distinguishing characteristics of CKD and are linked to patient death [7]. Numerous investigations have demonstrated that inflammation worsens renal function. White blood cell count, interleukin-6, his-CRP, and tumor necrosis factor-alpha receptor were found to have a favorable correlation with the outcome of CKD in a cross-sectional investigation by Shankar et al. [8]. Inflammation may be a sign of a poor prognosis in CKD patients, according to cohort research by Amdur RL et al. [9].

Due to their shared developmental ecological niche, the immune system and bone currently function as a tightly coupled functional unit (the bone immune system), with numerous anatomical and vascular sites of ongoing interaction between the two [10]. The role of the immune system in various skeletal pathologies is currently well established. Through direct or indirect influences on the physiological functions of bone cells, immune cells can eventually alter bone density [11, 12]. Some indices reflecting systemic immune and inflammatory status, such as the sex granulocyte-lymphocyte ratio [13], derived from immune cell counts, have also been correlated with BMD changes [14]. To better monitor the health of CKD patients, researchers are looking for novel indices based on immune cell counts to evaluate the risk of bone loss in patients.

A novel index that uses platelet, neutrophil, and lymphocyte counts called the Systemic Immune Inflammation Index (SII) can be used to measure the level of systemic inflammation [15, 16]. Growing data suggests that it gives insight into the body's overall immunological and inflammatory status and may be used to forecast risk and evaluate prognosis in conditions such as tumors, coronary artery disease, and bone loss [17–20]. Additionally, Qin et al. noted that higher SII was linked to a higher incidence of albuminuria in adults [21]. The association between BMD and SII in CKD patients is still unclear due to the small number of studies, and further research is required to determine the function of SII in the problems of bone loss in CKD patients.

The purpose of this study was to evaluate the association between SII and BMD in CKD patients and to assess the correlation between SII and the risk of bone loss/osteoporosis in CKD patients based on the theoretical backdrop discussed above. We hypothesized that increasing SII is linked to a higher risk of osteoporosis and that SII is inversely correlated with bone mineral density (BMD).

## Materials and methods

### Study subjects

The National Health and Nutrition Examination Survey (NHANES), which is based on a cross-sectional cohort study and intended to evaluate the nutritional and health status of the general population in the United States (US), served as the source for all subject data. Every two years, NHANES is updated and is associated with the US Centers for Disease Control and Prevention. Data from the NHANES 2011–2016 (2011–2012, 2013–2014, and 2015–2016) were retrieved. The authors do not have access to information that could identify individual participants during data collection.

Out of the 29,903 participants, we eliminated 5,324 people because their lymphocyte, neutrophil, and platelet counts were missing; 13,366 people because their blood creatinine, sex, race, and BMD measurements were missing; 8,541 people because they did not have CKD; 370 people because they were under 18; and 2,302 people because there were no pregnant people among the sample. The inclusion and exclusion details of the study population in this study are shown in **Fig 1.**

### Ethics statement

The NHANES procedure was approved by the National Center for Health Statistics Research Ethics Review Board, and written informed permission was acquired. The NHANES data were made publicly accessible after being anonymized. It allowed scholars to transform the data into a format that could be learned. To make sure that data were only utilized for statistical analyses and that all experiments were carried out in compliance with the relevant standards and laws, we agreed to abide by the study's data usage rules.

### SII calculations

Using automated hematology analytical tools, the lymphocyte, neutrophil, and platelet counts (expressed as 103 cells/L) were calculated. On the NHANES website, laboratory procedures for full blood count testing are described. Based on recent research [15], SII was calculated as PC * (NC / LC) for the platelet count (PC), neutrophil count (NC), and lymphocyte count (LC).

### Diagnosis of CKD

eGFR 90 ml/min/1.73 m$^2$ or eGFR >90 ml/min/1.73 m$^2$ and urine ACR >30 mg/L were both considered signs of CKD [22]. Both urinary ACR and serum creatinine were measured. Serum creatinine, ethnicity, and gender were used to compute the estimated glomerular filtration rate (eGFR) using the CKD-EPI equation [23]. The Kidney Disease Guidelines for Improving Global Outcomes [24] were used to identify the stage of CKD.

### Acquisition of bone density

In actuality, no one location or method can completely satisfy the therapeutic demands for bone mineral assessments. The evaluation of osteoporosis can be impacted by biological variations in bone composition that exist between populations and locations, as well as by

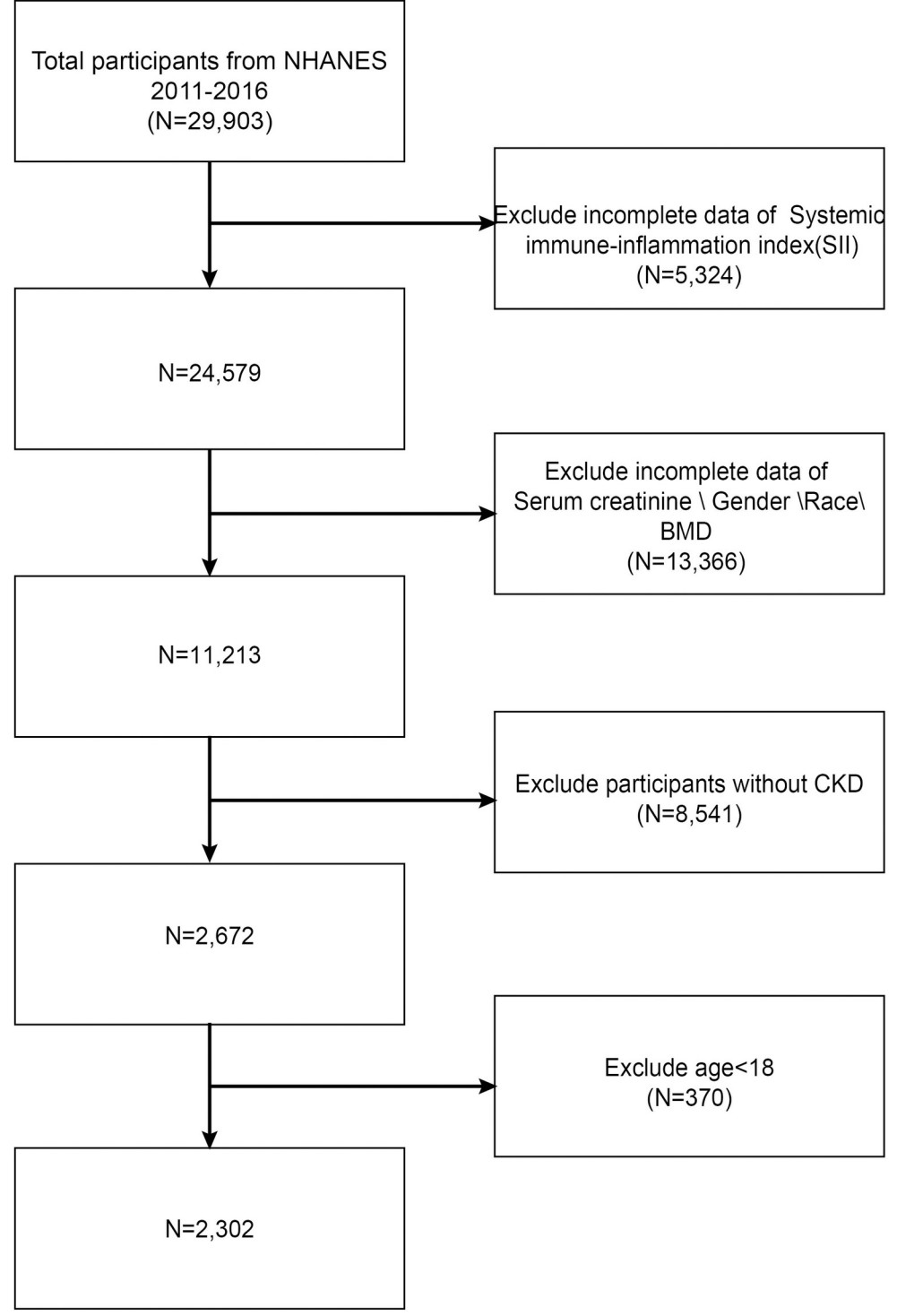

**Fig 1. The flow of participants through the study.**

technological inaccuracies in the accuracy of various measures [25]. The most used bone densitometry technology, DXA, is used to measure the mineral content of bone at several places across the skeleton, particularly those that are most likely to fracture [26]. Hip fractures are a

serious consequence of osteoporosis in clinical terms [27]. The World Health Organization defines osteoporosis as having a femoral neck DXA score that is below normal [25]. The majority of bone loss in CKD patients comes from the cortical bone [28], and hip fracture risk is higher in ESRD and milder stages of CKD [29]. The frequency of poor BMD, however, varied from 50 to 80 percent in the radius, 16 to 47 percent in the femoral neck, and 13 to 29 percent in the lumbar spine in many investigations on hemodialysis patients [30]. In order to better link BMD with inflammation at various places, we included BMD of the pelvis, left upper and lower limbs, and lumbar spine in the current investigation. We also included BMD of the thoracic spine and trunk in the study.

A qualified radiographer used a Hologic QDR-4500A fan-beam densitometer (Hologic; Bedford, MA, USA) to perform dual-energy X-ray absorptiometry (DXA) exams on all participants, including those who were included in the final analyses. Using Hologic APEX software (version 4.0), all DXA examination data were analyzed. The NHANES website offers further details.

## Covariates

This study additionally included covariates in the analysis to account for the possible impact of other variables on bone metabolism. Age, race, body mass index (BMI), smoking, hypertension, diabetes mellitus, blood alkaline phosphatase, serum uric acid, blood calcium, blood phosphorus, blood vitamin D, and blood triglycerides were eventually added as covariates based on other research [31, 32]. In this instance, the final blood pressure reading was calculated as the average of the body measures, and hypertension was defined as a mean systolic blood pressure of greater than 140 mmHg and/or a mean diastolic blood pressure of greater than 90 mmHg. Diabetes mellitus was deemed to exist when the hemoglobin A1c (HbA1c) level reached 6.5%. How these variables were estimated is fully described on the NHANES website (https://www.cdc.gov/nchs/nhanes/).

## Analyses of statistics

With statistical significance set at P 0.05, we conducted all statistical analyses using R (http://www.r-project.org) and EmpowerStats (http://www.empowerstats.com). Model 1 included no covariate adjustments, Model 2 had ethnicity, age, and gender adjustments, and Model 3 included all of the covariate adjustments from Table 1. Additionally, subgroup analyses were performed. To deal with nonlinearities, the Generalised Additive Model (GAM) and smooth curve fitting were employed.

## Results

### Baseline characteristics of the SII-strategized population

2,302 people were examined in this study. Depending on their SII levels, the research sample was evenly split into Qs (Q1-Q4). Table 1 displays weighted demographic and medical information. The study had 2,302 adult participants in total. The Qs were statistically different (p < 0.05) for urinary albumin, blood alkaline phosphatase, blood phosphorus, blood uric acid, blood creatinine, blood cholesterol, blood urea nitrogen, glomerular filtration rate, bone density in the left arm, bone density in the left leg, bone density in the thoracic vertebrae, sex, ethnicity, whether or not they were smokers, BMI, and diabetes mellitus, whereas the age of There were no statistically significant differences in comparisons of total calcium, blood vitamin D, urinary creatinine, urinary albumin-creatinine ratio, lumbar spine bone density, pelvic bone density, trunk bone density, and hypertension. Additionally, we noticed that patients in

**Table 1. Description of 2,302 participants included in the present study.**

| Characteristics | Q1 | Q2 | Q3 | Q4 | P value |
|---|---|---|---|---|---|
| N | 576 | 573 | 577 | 576 | |
| Age (years) | 44.488 ± 11.117 | 44.412 ± 10.447 | 44.560 ± 10.922 | 44.724 ± 11.003 | 0.968 |
| Total Calcium (mg/dL) | 2.346 ± 0.091 | 2.356 ± 0.088 | 2.354 ± 0.086 | 2.359 ± 0.098 | 0.096 |
| 25OHD2+25OHD3 (nmol/L) | 63.203 ± 28.906 | 65.243 ± 27.110 | 64.057 ± 25.920 | 66.087 ± 25.829 | 0.279 |
| 25OHD2 (nmol/L) | 3.596 ± 10.980 | 3.557 ± 11.726 | 4.122 ± 12.854 | 3.783 ± 11.338 | 0.841 |
| Albumin, urine (ug/mL) | 72.630 ± 250.402 | 73.659 ± 357.259 | 88.322 ± 381.993 | 138.666 ± 667.685 | 0.038 |
| Creatinine, urine (mg/dL) | 131.101 ± 88.452 | 133.274 ± 83.573 | 136.662 ± 85.280 | 140.875 ± 90.268 | 0.250 |
| Albumin creatinine ratio (mg/g) | 66.559 ± 256.037 | 67.308 ± 286.152 | 80.224 ± 449.240 | 123.264 ± 610.823 | 0.081 |
| Alkaline Phosphatase (ALP) (IU/L) | 66.446 ± 23.509 | 65.461 ± 19.062 | 67.557 ± 21.268 | 69.542 ± 22.202 | 0.010 |
| Phosphorus (mg/dL) | 1.204 ± 0.184 | 1.223 ± 0.184 | 1.236 ± 0.183 | 1.223 ± 0.204 | 0.032 |
| Uric acid (mg/dL) | 323.144 ± 84.056 | 338.505 ± 90.126 | 344.375 ± 84.532 | 354.352 ± 89.122 | <0.001 |
| Creatinine (mg/dL) | 0.978 ± 0.421 | 1.014 ± 0.509 | 1.001 ± 0.376 | 1.114 ± 0.912 | <0.001 |
| Cholesterol (mmol/L) | 5.209 ± 1.136 | 5.108 ± 1.095 | 5.098 ± 1.088 | 5.024 ± 1.110 | 0.043 |
| Blood Urea Nitrogen (mg/dL) | 13.036 ± 5.102 | 13.736 ± 5.135 | 13.865 ± 5.620 | 14.705 ± 7.611 | <0.001 |
| GFR (mL/min/1.73m$^2$) | 88.430 ± 20.421 | 85.214 ± 17.720 | 85.691 ± 18.615 | 83.101 ± 21.066 | <0.001 |
| Left Arm BMD (g/cm$^2$) | 0.767 ± 0.103 | 0.788 ± 0.098 | 0.785 ± 0.106 | 0.792 ± 0.101 | <0.001 |
| Left Leg BMD (g/cm$^2$) | 1.161 ± 0.152 | 1.189 ± 0.145 | 1.174 ± 0.139 | 1.189 ± 0.140 | 0.001 |
| Thoracic Spine BMD (g/cm$^2$) | 0.822 ± 0.130 | 0.841 ± 0.123 | 0.827 ± 0.121 | 0.843 ± 0.132 | 0.013 |
| Lumbar Spine BMD (g/cm$^2$) | 1.043 ± 0.161 | 1.051 ± 0.157 | 1.036 ± 0.149 | 1.044 ± 0.167 | 0.477 |
| Pelvis BMD (g/cm$^2$) | 1.247 ± 0.173 | 1.263 ± 0.163 | 1.256 ± 0.169 | 1.244 ± 0.167 | 0.235 |
| Trunk Bone BMD (g/cm$^2$) | 0.895 ± 0.122 | 0.906 ± 0.113 | 0.894 ± 0.114 | 0.896 ± 0.115 | 0.256 |
| Gender (%) | | | | | <0.001 |
| Men | 241 (41.840%) | 300 (52.356%) | 296 (51.300%) | 347 (60.243%) | |
| Women | 335 (58.160%) | 273 (47.644%) | 281 (48.700%) | 229 (39.757%) | |
| Race/Ethnicity (%) | | | | | <0.001 |
| Mexican American | 61 (10.590%) | 60 (10.471%) | 72 (12.478%) | 59 (10.243%) | |
| Other Hispanic | 43 (7.465%) | 54 (9.424%) | 68 (11.785%) | 58 (10.069%) | |
| Non-Hispanic White | 158 (27.431%) | 247 (43.106%) | 260 (45.061%) | 309 (53.646%) | |
| Non-Hispanic Black | 205 (35.590%) | 116 (20.244%) | 109 (18.891%) | 95 (16.493%) | |
| Non-Hispanic Asian | 82 (14.236%) | 63 (10.995%) | 48 (8.319%) | 33 (5.729%) | |
| Other Race—Including Multi-Racial | 27 (4.688%) | 33 (5.759%) | 20 (3.466%) | 22 (3.819%) | |
| Hypertension, n (%) | | | | | 0.799 |
| No | 482 (85.159%) | 482 (85.765%) | 472 (83.986%) | 473 (84.014%) | |
| Yes | 84 (14.841%) | 80 (14.235%) | 90 (16.014%) | 90 (15.986%) | |
| Smoke now recoded | | | | | <0.001 |
| No | 103 (17.882%) | 128 (22.339%) | 101 (17.504%) | 121 (21.007%) | |
| Yes | 102 (17.708%) | 115 (20.070%) | 131 (22.704%) | 174 (30.208%) | |
| Unknown | 371 (64.410%) | 330 (57.592%) | 345 (59.792%) | 281 (48.785%) | |
| BMI (kg/m$^2$) | | | | | <0.001 |
| < = 25 | 193 (33.682%) | 148 (25.829%) | 132 (23.037%) | 134 (23.304%) | |
| >25, < = 30 | 181 (31.588%) | 184 (32.112%) | 186 (32.461%) | 184 (32.000%) | |
| >30 | 199 (34.729%) | 241 (42.059%) | 255 (44.503%) | 257 (44.696%) | |
| Diabetes, n (%) | | | | | 0.033 |
| No | 521 (90.451%) | 510 (89.161%) | 507 (87.868%) | 490 (85.069%) | |
| Yes | 55 (9.549%) | 62 (10.839%) | 70 (12.132%) | 86 (14.931%) | |

BMI, body mass index; GFR, glomerular filtration rate. Mean ± sd. for continuous variables: P value was calculated using a weighted linear regression model. % for Categorical variables: P value was calculated by weighted chi-square test.

the first quartile of SII had greater blood cholesterol and glomerular filtration rates compared to other subgroups, but lower blood creatinine and blood urea nitrogen.

The GFR values for the CKD patients in this study were computed using the CKD-EPI algorithm, and the staging of the participating patients was done in accordance with the KDIGO recommendations: In terms of lumbar spine bone mineral density and current smoking status across CKD subgroups, there was no statistically significant difference between individuals with different CKD stages (p> 0.05). The sample was mostly focused on CKD stage 2, and SII tended to rise as CKD advanced, despite statistically significant differences between the remaining variables. The result is shown in Table 2.

## Relationship between SII and BMD in CKD patients

Table 3 displays the outcomes of the multivariate regression analysis. We initially created an unadjusted model to investigate the relationship between SII and BMD at various locations. We discovered that the SII percentile was significantly linked with BMD in the left arm alone (0.006, 95% CI: 0.002, 0.009, p = 0.00370). However, after controlling for sex, age, and race in adjusted model I, SII was associated with pelvic BMD, suggesting that a higher SII percentile was associated with lower odds of pelvic BMD. This association also suggested that the higher SII percentile was associated with a trend towards decreasing pelvic BMD as the SII percentile increased (trend p = 0.01125). Age, sex, race, blood alkaline phosphatase, blood urea nitrogen, blood calcium, blood triglycerides, blood phosphorus, blood creatinine, blood uric acid, total active vitamin D, 2,5-hydroxyvitamin D2, presence of hypertension, diabetes mellitus, BMI, and calculated GFR were all taken into account in the adjusted Model II. Further analyses for pelvic BMD and SII were then carried out, and only pelvic BMD remained substantially linked with SII.

**Fig 2** displays the smoothed curve fits and scatter plots. Model III was used to fit a smoothed curve to represent the nonlinear connection between SII and PEBMD. There were no statistically significant optimum breakpoints identified using a two-stage linear regression model. The detailed results of the threshold effect analysis are presented in Table 4. Overall, SII and PEBMD have a negative correlation, as can be observed.

## SII and pelvic bone density in CKD patients: Stratified analysis and investigation of the threshold effect

In order to analyze changes in pelvic BMD caused by SII in patients with CKD, the sample was stratified using the previously reported adjusted model. Because there weren't enough stage 5 patients with CKD, stage 4 and stage 5 patients were analyzed together:

Our findings indicated that the negative connection between SII and pelvic BMD was independently and significantly positive in men [0.0008 (-0.023, -0.006)] but not statistically significant in the female models when subgroup analyses stratified by sex were performed. Independent statistical significance for different racial groups was absent. Patients without diabetes, hypertension, or a BMI of more than 30 kg/m$^2$ demonstrated a substantial positive connection between SII and pelvic BMD rather than the expected negative correlation. Patients with CKD stage 2 showed a negative connection between SII and pelvic BMD when we grouped the patients based on CKD stage [0.0222 (-0.015, -0.001)]. The results are shown in Table 5.

**Fig 3** displays the outcomes of further stratified analyses based on patient gender, the presence of diabetes, hypertension, and a BMI greater than 30 kg/m$^2$, as well as smoothed curve fitting using model 3.

**Table 2. Description of 2,302 participants included in the present study.**

| Characteristics | 1 | 2 | 3 | 4 | 5 | P-value |
|---|---|---|---|---|---|---|
| N | 510 | 1673 | 97 | 9 | 13 | |
| Age (years) | 39.492 ± 11.845 | 45.751 ± 10.178 | 49.835 ± 8.701 | 44.556 ± 7.333 | 48.231 ± 9.968 | <0.001 |
| Total Calcium (mg/dL) | 2.338 ± 0.089 | 2.359 ± 0.087 | 2.368 ± 0.128 | 2.278 ± 0.109 | 2.323 ± 0.158 | <0.001 |
| 25OHD2+25OHD3 (nmol/L) | 53.294 ± 21.893 | 67.748 ± 27.198 | 71.280 ± 29.665 | 63.700 ± 42.276 | 61.969 ± 26.606 | <0.001 |
| 25OHD2 (nmol/L) | 2.934 ± 7.205 | 3.632 ± 11.922 | 6.009 ± 15.799 | 20.719 ± 28.458 | 24.976 ± 33.351 | <0.001 |
| Alkaline Phosphatase (ALP) (IU/L) | 70.661 ± 22.296 | 65.589 ± 20.425 | 73.155 ± 24.410 | 65.000 ± 18.432 | 105.231 ± 51.357 | <0.001 |
| Phosphorus (mg/dL) | 1.214 ± 0.185 | 1.218 ± 0.184 | 1.245 ± 0.198 | 1.392 ± 0.133 | 1.684 ± 0.352 | <0.001 |
| Uric acid (mg/dL) | 312.226 ± 86.494 | 343.923 ± 81.646 | 402.994 ± 99.732 | 535.300 ± 231.601 | 336.754 ± 129.381 | <0.001 |
| Creatinine (mg/dL) | 0.723 ± 0.154 | 1.036 ± 0.161 | 1.445 ± 0.337 | 2.838 ± 0.685 | 7.365 ± 3.291 | <0.001 |
| Cholesterol (mmol/L) | 5.060 ± 1.195 | 5.130 ± 1.067 | 5.137 ± 1.259 | 5.138 ± 1.687 | 4.215 ± 0.983 | 0.038 |
| Blood Urea Nitrogen (mg/dL) | 11.316 ± 3.810 | 13.805 ± 4.145 | 20.660 ± 10.724 | 47.000 ± 20.609 | 42.769 ± 16.513 | <0.001 |
| GFR (mL/min/1.73m$^2$) | 113.434 ± 13.559 | 80.013 ± 8.254 | 51.928 ± 7.690 | 22.997 ± 4.006 | 8.912 ± 3.087 | <0.001 |
| Albumin, urine (ug/mL) | 165.779 ± 490.847 | 27.149 ± 190.702 | 217.935 ± 1015.095 | 1214.790 ± 2029.890 | 983.027 ± 829.289 | <0.00001 |
| Creatinine, urine (mg/dL) | 113.376 ± 77.705 | 133.638 ± 84.815 | 114.027 ± 82.924 | 85.810 ± 39.787 | 69.360 ± 30.377 | 0.00003 |
| Albumin creatinine ratio (mg/g) | 147.080 ± 390.210 | 23.879 ± 194.352 | 148.087 ± 603.868 | 1678.315 ± 2989.744 | 1403.805 ± 978.625 | <0.00001 |
| Left Arm BMD (g/cm$^2$) | 0.755 ± 0.097 | 0.793 ± 0.102 | 0.766 ± 0.103 | 0.804 ± 0.083 | 0.712 ± 0.123 | <0.001 |
| Left Leg BMD (g/cm$^2$) | 1.148 ± 0.142 | 1.190 ± 0.144 | 1.145 ± 0.127 | 1.207 ± 0.151 | 1.081 ± 0.222 | <0.001 |
| Thoracic Spine BMD (g/cm$^2$) | 0.825 ± 0.128 | 0.836 ± 0.125 | 0.821 ± 0.136 | 0.930 ± 0.189 | 0.824 ± 0.156 | 0.049 |
| Lumbar Spine BMD (g/cm$^2$) | 1.036 ± 0.156 | 1.047 ± 0.158 | 1.025 ± 0.157 | 1.121 ± 0.204 | 1.055 ± 0.208 | 0.252 |
| Pelvis BMD (g/cm$^2$) | 1.224 ± 0.162 | 1.265 ± 0.166 | 1.201 ± 0.179 | 1.382 ± 0.312 | 1.088 ± 0.139 | <0.001 |
| Trunk Bone BMD (g/cm$^2$) | 0.882 ± 0.112 | 0.905 ± 0.116 | 0.864 ± 0.111 | 0.978 ± 0.181 | 0.822 ± 0.126 | <0.001 |
| SII | 1.936 ± 1.042 | 2.022 ± 1.118 | 2.347 ± 1.173 | 2.494 ± 1.496 | 2.541 ± 1.248 | 0.003 |
| Gender (%) | | | | | | <0.001 |
| Men | 197 (38.627%) | 933 (55.768%) | 46 (47.423%) | 2 (22.222%) | 6 (46.154%) | |
| Women | 313 (61.373%) | 740 (44.232%) | 51 (52.577%) | 7 (77.778%) | 7 (53.846%) | |
| Race/Ethnicity (%) | | | | | | <0.001 |
| Mexican American | 100 (19.608%) | 139 (8.308%) | 8 (8.247%) | 4 (44.444%) | 1 (7.692%) | |
| Other Hispanic | 60 (11.765%) | 149 (8.906%) | 13 (13.402%) | 1 (11.111%) | 0 (0.000%) | |
| Non-Hispanic White | 141 (27.647%) | 791 (47.280%) | 42 (43.299%) | 0 (0.000%) | 0 (0.000%) | |
| Non-Hispanic Black | 129 (25.294%) | 360 (21.518%) | 21 (21.649%) | 4 (44.444%) | 11 (84.615%) | |
| Non-Hispanic Asian | 59 (11.569%) | 159 (9.504%) | 7 (7.216%) | 0 (0.000%) | 1 (7.692%) | |
| Other Race—Including Multi-Racial | 21 (4.118%) | 75 (4.483%) | 6 (6.186%) | 0 (0.000%) | 0 (0.000%) | |
| Hypertension, n (%) | | | | | | <0.001 |
| No | 384 (77.264%) | 1431 (87.203%) | 80 (86.022%) | 5 (55.556%) | 9 (69.231%) | |
| Yes | 113 (22.736%) | 210 (12.797%) | 13 (13.978%) | 4 (44.444%) | 4 (30.769%) | |
| Smoke now recoded | | | | | | 0.253 |
| No | 86 (40.952%) | 338 (47.740%) | 22 (47.826%) | 3 (50.000%) | 4 (80.000%) | |
| Yes | 124 (59.048%) | 370 (52.260%) | 24 (52.174%) | 3 (50.000%) | 1 (20.000%) | |
| BMI (kg/m$^2$) | | | | | | <0.001 |
| < = 25 | 152 (30.159%) | 434 (25.972%) | 17 (17.526%) | 0 (0.000%) | 4 (30.769%) | |
| >25, < = 30 | 108 (21.429%) | 594 (35.548%) | 28 (28.866%) | 0 (0.000%) | 5 (38.462%) | |
| >30 | 244 (48.413%) | 643 (38.480%) | 52 (53.608%) | 9 (100.000%) | 4 (30.769%) | |
| Diabetes, n (%) | | | | | | <0.001 |
| No | 403 (79.020%) | 1536 (91.866%) | 74 (76.289%) | 5 (55.556%) | 10 (76.923%) | |
| Yes | 107 (20.980%) | 136 (8.134%) | 23 (23.711%) | 4 (44.444%) | 3 (23.077%) | |

BMI, body mass index; GFR, glomerular filtration rate. Mean ± sd. for continuous variables: P value was calculated using a weighted linear regression model. % for Categorical variables: P value was calculated by weighted chi-square test.

**Table 3. Association of SII with GFR among 2,302 CKD patients, NHANES 2011–2016.**

| Characteristics | Model 1 | Model 2 | Model 3 |
|---|---|---|---|
| | β (95% CI) P value [a] | β (95% CI) P value [b] | β (95% CI) P value [c] |
| Left Arm BMD (g/ cm$^2$) | 0.006 (0.002, 0.009) 0.00370 | -0.001 (-0.004, 0.001) 0.28068 | -0.001 (-0.004, 0.001) 0.36145 |
| Left Leg BMD (g/ cm$^2$) | 0.005 (-0.000, 0.010) 0.07565 | -0.002 (-0.006, 0.002) 0.33328 | -0.002 (-0.006, 0.002) 0.38476 |
| Thoracic Spine BMD (g/ cm$^2$) | 0.002 (-0.002, 0.007) 0.32539 | 0.000 (-0.004, 0.005) 0.91423 | -0.001 (-0.006, 0.003) 0.58741 |
| Lumbar Spine BMD (g/ cm$^2$) | -0.000 (-0.006, 0.005) 0.87219 | 0.001 (-0.005, 0.006) 0.85132 | 0.001 (-0.005, 0.007) 0.77892 |
| Pelvis BMD (g/ cm$^2$) | -0.005 (-0.011, 0.001) 0.12507 | -0.008 (-0.014, -0.002) 0.01125 | -0.006 (-0.012, -0.000) 0.03368 |

[a] Model 1, no covariates were adjusted.

[b] Model 2, Adjust for sex, age, and race.

[c] Model 3. Adjust for Age, sex, race, blood alkaline phosphatase, blood urea nitrogen, blood calcium, blood triglycerides, blood phosphorus, blood creatinine, blood uric acid, total active vitamin D, 2,5-hydroxyvitamin D2, presence of hypertension, diabetes mellitus, BMI, and calculated GFR. Generalized additive models were applied.

The relationship between SII and PEBMD showed a significant inverted U-shaped curve with a fold point of 0.969 ((1,000 cells/l)) after stratifying the sample for patients who did not have diabetes mellitus, according to a threshold effect analysis using a two-stage linear regression model. The connection between SII and PEBMD likewise displayed an inverted U-shaped curve in individuals without and with hypertension, with fold points of 0.938 (1,000 cells/l) and 2.947 (1,000 cells/l), respectively. SII and PEBMD had a more complicated non-linear association in individuals with BMI more than 30 kg/m$^2$, with no significant fold points. Tables 6 and 7 show the findings of the threshold effect investigation.

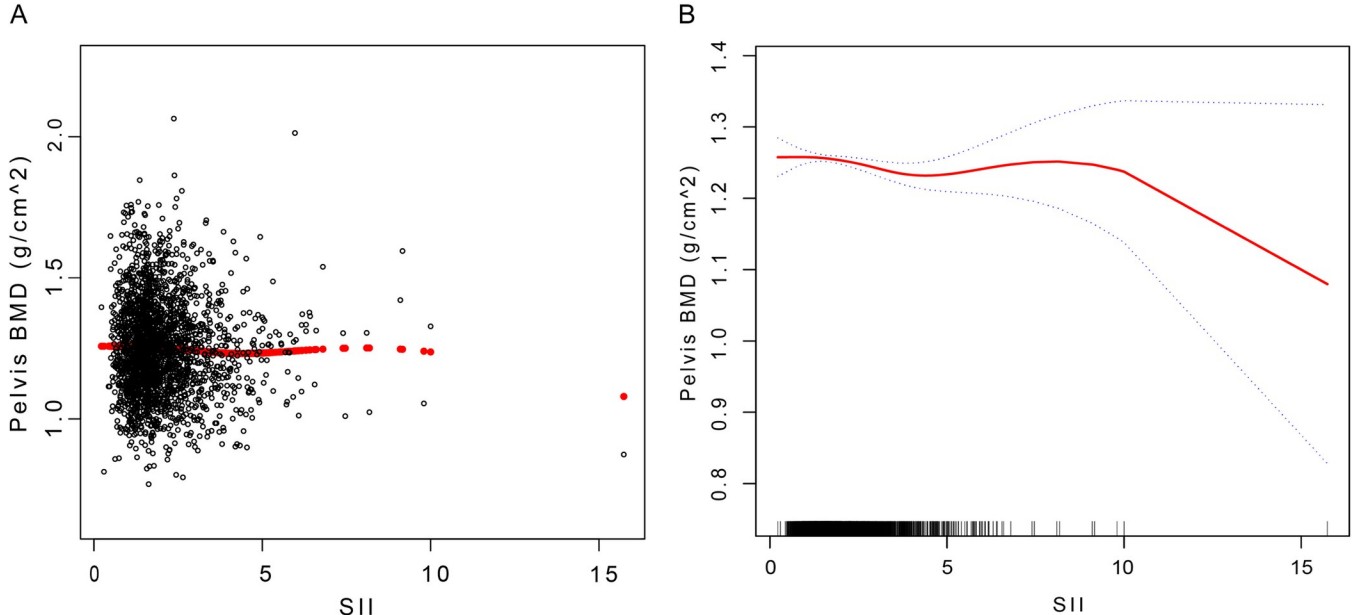

**Fig 2. Association between SII and Pelvis BMD.** (A) Each black dot represents a sample. (B) The solid red line represents a smooth curve fit between the variables. Blue bands indicate 95% confidence intervals of the fit. SII, systemic immunoinflammatory index; PEBMD, pelvic bone mineral density.

**Table 4. Threshold effect analysis of SII and pelvic BMD in CKD patients using Model 3.**

| Outcome [b] | Pelvis BMD (g/ cm$^2$) [a] |
|---|---|
| Model I | |
| one-line effect | -0.006 (-0.012, -0.000) 0.0337 |
| Model II | |
| Folding point (K) | 0.912 |
| < K-segment effect 1 | 0.096 (-0.022, 0.215) 0.1100 |
| > K-segment effect 2 | -0.007 (-0.013, -0.002) 0.0136 |
| The difference in effect between 2 and 1 | -0.104 (-0.223, 0.016) 0.0885 |
| Predicted value of the equation at the breakpoint | 1.260 (1.249, 1.270) |
| Log-likelihood ratio test | 0.086 |
| 95 confidence interval at the breakpoint | 0.838, 1.15 |

[a] Table data: β (95%CI) P value / OR (95%CI) P value

[b] Exposure variable: SII

Adjustment variable. Age, sex, race, blood alkaline phosphatase, blood urea nitrogen, blood calcium, blood triglycerides, blood phosphorus, blood creatinine, blood uric acid, total active vitamin D, 2,5-hydroxyvitamin D2, presence of hypertension, diabetes mellitus, BMI, and calculated GFR

Some hints are given by the variations in the association between SII and PEBMD in the various subgroups, but more research is required to validate these correlations and dive further into the underlying processes.

## Discussion

Long recognized as a prognostic factor in a variety of illnesses, the immune system. According to recent findings from cross-sectional studies of the NHANES-III 2011–2016 database, SII levels in individuals with CKD correlate with pelvic BMD, which declines with rising SII.

To our knowledge, this is the first community-based, nationally representative cohort of people from the United States to evaluate the relationship between SII, a less expensive clinical measure, and reduced bone density, a typical consequence of CKD.

We discovered that SII levels were negatively correlated with pelvic BMD in patients with CKD by comparing regression models after unadjusted modeling, the addition of demographic variables, and the remaining pertinent variables. This finding corroborated the persistent finding that hip BMD can be used to predict fractures, as mentioned in the KDIGO guidelines [33]. Additionally, when examining the relationship between SII and PEBMD, the findings of stratified analyses revealed that men who were obese, male, and free of diabetes or hypertension had a stronger negative association between SII and PEBMD. These might serve as some benchmarks for further clinical choices.

According to a 2014 study by Hu [15] et al., SII is a thorough and unique inflammatory biomarker. Prior research has routinely employed SII as a predictor of the development of kidney transplant rejection as well as the incidence of acute or chronic renal injury [34–38]. Numerous studies connected to SII have been conducted in recent years, and they have thoroughly examined its clinical importance. Lai et al. [34] in a cohort study showed that elevated levels of SII prior to CAG were an important and independent risk factor for postoperative AKI, and Halpern et al. [39] in a single-center cohort study showed that elevated levels of SII were independently associated with increased survival only in post-transplant patients. For instance, Xie et al. [40] in 2022 found that elevated levels of SII were associated with hepatic steatosis but not with hepatic fibrosis. A cohort study was conducted by Halpern et al. in one center.

**Table 5. Stratified analysis of SII and pelvic BMD in patients with CKD.**

| Subgroups | N | Model 2 [a] | Model 3 [b] |
|---|---|---|---|
| | | β (95% CI) P value | β (95%CI) P value |
| Gender (%) | | | |
| Men | 1184 | | -0.014 (-0.023, -0.006) 0.0008 |
| Women | 1118 | | -0.002 (-0.010, 0.007) 0.6559 |
| Race/Ethnicity (%) | | | |
| Mexican American | 252 | | -0.010 (-0.026, 0.005) 0.2012 |
| Other Hispanic | 223 | | 0.002 (-0.019, 0.022) 0.8611 |
| Non-Hispanic White | 974 | | -0.008 (-0.017, 0.001) 0.0845 |
| Non-Hispanic Black | 525 | | 0.002 (-0.011, 0.015) 0.7369 |
| Non-Hispanic Asian | 226 | | -0.015 (-0.040, 0.011) 0.2592 |
| Other Race—Including Multi-Racial | 102 | | -0.017 (-0.057, 0.024) 0.4151 |
| Diabetes, n (%) | | | |
| No | 2028 | -0.009 (-0.015, -0.003) 0.0051 | -0.006 (-0.013, -0.000) 0.0407 |
| Yes | 273 | 0.001 (-0.019, 0.020) 0.9394 | -0.005 (-0.026, 0.016) 0.6200 |
| Hypertension, n (%) | | | |
| No | 1909 | -0.008 (-0.015, -0.002) 0.0100 | -0.007 (-0.013, -0.000) 0.0387 |
| Yes | 344 | -0.007 (-0.024, 0.010) 0.4249 | -0.009 (-0.026, 0.008) 0.3240 |
| CKDS Stages | | | |
| 1 | 510 | -0.003 (-0.017, 0.010) 0.6223 | -0.002 (-0.015, 0.011) 0.7798 |
| 2 | 1673 | -0.009 (-0.016, -0.003) 0.0071 | -0.008 (-0.015, -0.001) 0.0222 |
| 3 | 97 | -0.005 (-0.035, 0.024) 0.7180 | 0.004 (-0.026, 0.033) 0.8110 |
| 4+5 | 22 | 0.067 (-0.036, 0.170) 0.2202 | 0.040 (-0.137, 0.217) 0.6789 |
| BMI (kg/m$^2$) | | | |
| < = 25 | 607 | -0.006 (-0.016, 0.005) 0.2675 | -0.004 (-0.014, 0.007) 0.4796 |
| >25, < = 30 | 735 | -0.001 (-0.011, 0.009) 0.8264 | 0.001 (-0.009, 0.011) 0.9030 |
| >30 | 952 | -0.015 (-0.024, -0.006) 0.0017 | -0.014 (-0.024, -0.005) 0.0031 |

[a] Model 2, Adjust for sex, age, and race.

[b] Model 3. Adjust for Age, sex, race, blood alkaline phosphatase, blood urea nitrogen, blood calcium, blood triglycerides, blood phosphorus, blood creatinine, blood uric acid, total active vitamin D, 2,5-hydroxyvitamin D2, presence of hypertension, diabetes mellitus, BMI, and calculated GFR. Generalized additive models were applied.

Regarding prognosis, prospective cohort research by Shi [31] et al. showed that in CKD patients with ACS, higher SII was linked to poor cardiovascular outcomes. In a similar vein, Xie et al. [41] discovered a favorable correlation between SII and abdominal aortic calcification, a frequent CKD consequence. In a cohort investigation of critically ill AKI patients, Lan et al. [42] discovered a J-shaped relationship between SII and all-cause death in these patients.

Studying the onset of CKD's consequences, however, can be more successful in enhancing the quality of patients' survival because CKD is a long-term chronic disease. Long-term, chronic inflammation can also exacerbate renal anemia and renal function [43], as well as promote malnutrition, in the kidney, the site of the majority of renal disorders [44]. To preserve the kidneys, the body suppresses inflammation through autophagy [45]. The very diverse etiology of bone disease, as well as the restrictions and particular adverse effects of available treatments, make it difficult to diagnose and treat osteoporosis in individuals with severe CKD [46]. Therefore, it is crucial to identify osteoporosis early and take steps to avoid it. We opted to investigate inflammatory markers, a mechanism frequently present in BMD decrease and nephropathy, and ultimately found specific outcomes.

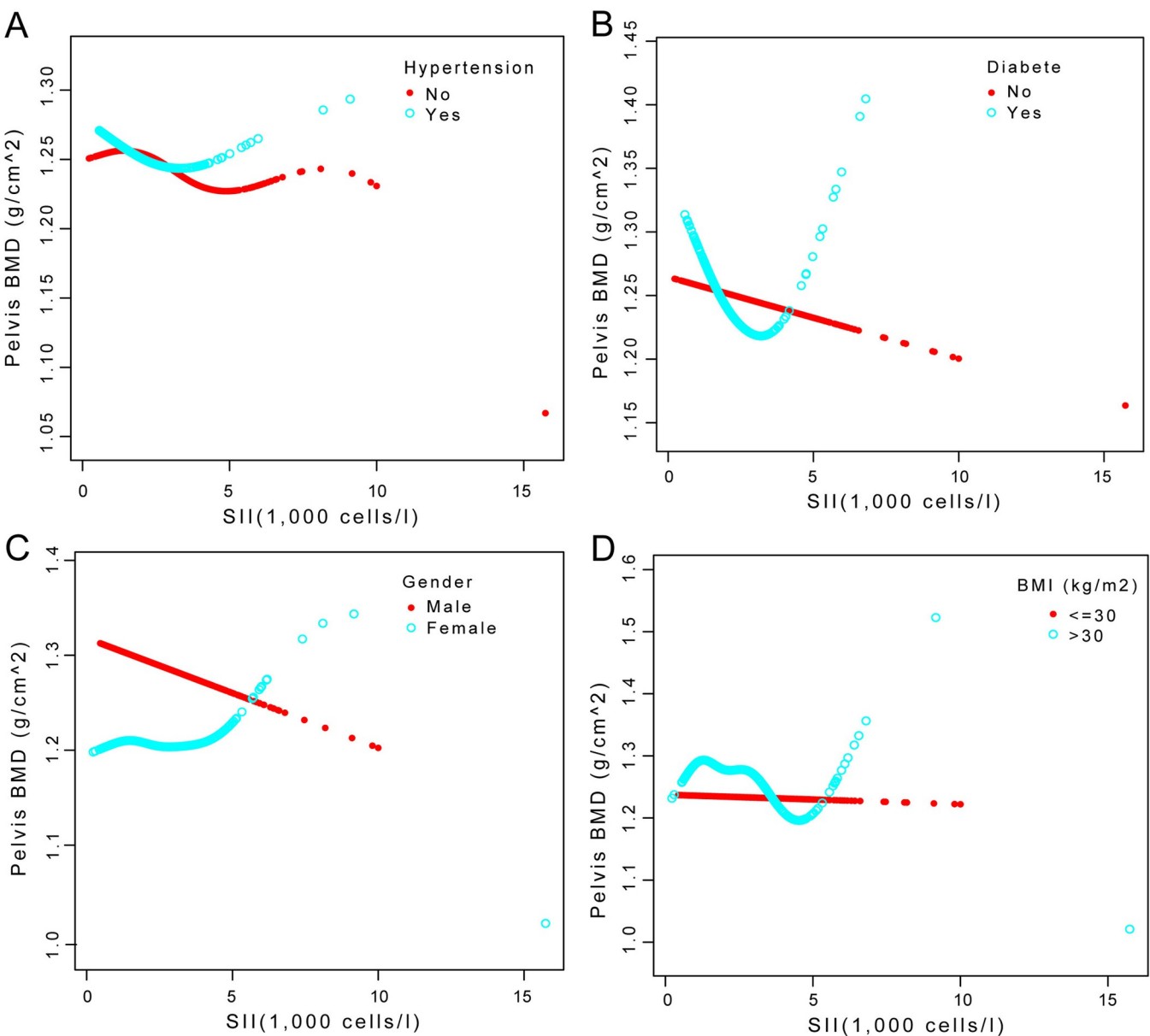

**Fig 3. Model 3 dictated the stratification of the study's participants based on gender, the existence of hypertension, the presence of diabetes, and a BMI of more than 30 kg/m², after which smoothed curves fitting the model were shown.**

Our research has a few drawbacks. Firstly, since this research is cross-sectional, temporality cannot be established. Additionally, even after controlling for a number of pertinent confounders, we were unable to completely exclude the impact of other confounders, therefore it is important to proceed with care when interpreting our results. Third, while patients with CKD typically use oral drugs like hormones depending on the underlying condition, our findings could not accurately reflect the true situation because the NHANES database's constraints prevented us from including individuals' medication use as a covariate. Fourth, although the CKD-EPI equation is the most accurate GFR estimation equation that has been tested in a wide range of populations and is appropriate for general clinical use [47], its accuracy is still

**Table 6. Using Model 3, SII and pelvic BMD in CKD patients without diabetes were analyzed for threshold effects after smoothed curve fitting.**

| Outcome: | Pelvis BMD (g/cm$^2$) |
|---|---|
| Model I | |
| one-line effect | -0.005 (-0.011, 0.001) 0.1207 |
| Model II | |
| Folding point (K) | 0.969 |
| < K-segment effect 1 | 0.113 (0.008, 0.218) 0.0355 |
| > K-segment effect 2 | -0.007 (-0.013, -0.000) 0.0384 |
| The difference in effect between 2 and 1 | -0.120 (-0.226, -0.013) 0.0279 |
| Predicted value of the equation at the breakpoint | 1.261 (1.250, 1.272) |
| Log-likelihood ratio test | 0.027 |
| 95 confidence interval at the breakpoint | 0.833, 1.194 |

not comparable to filtered marker measurements. The degree of renal impairment in this study was extrapolated from the CKD-EPI equation.

Despite these drawbacks, our study provides a number of advantages. Our study is typical of the multiracial and gender-diverse adult population of the United States since we employed a nationally representative sample. Furthermore, the size of the sample used in our study allowed us to do a subgroup analysis.

## Conclusion

In our study, pelvic bone density in CKD patients was associated with SII levels, and pelvic bone density reduced as SII increased. The change in BMD with SII in CKD patients without diabetes may be at a turning point. SII and pelvic bone density were substantially associated with individuals with stage 2 CKD. This may offer recommendations for the prevention and management of problems from osteoporosis in CKD patients. To support our findings, further comprehensive prospective cohort studies are still required.

**Table 7. Using Model 3, CKD patients were stratified according to whether they had hypertension or not, followed by a threshold effect analysis of their SII and pelvic BMD, respectively, after smoothing curve fitting.**

| Outcome | Pelvis BMD (g/ cm$^2$) | |
|---|---|---|
| | No hypertension | Hypertension |
| Model I | | |
| one-line effect | -0.006 (-0.013, -0.000) 0.0386 | -0.008 (-0.025, 0.008) 0.3129 |
| Model II | | |
| Folding point (K) | 0.938 | 2.947 |
| < K-segment effect 1 | 0.107 (-0.006, 0.220) 0.0637 | -0.032 (-0.061, -0.003) 0.0291 |
| > K-segment effect 2 | -0.008 (-0.014, -0.002) 0.0129 | 0.019 (-0.013, 0.050) 0.2494 |
| The difference in effect between 2 and 1 | -0.115 (-0.230, -0.001) 0.0490 | 0.051 (0.000, 0.102) 0.0508 |
| Predicted value of the equation at the breakpoint | 1.261 (1.250, 1.272) | 1.237 (1.196, 1.278) |
| Log-likelihood ratio test | 0.047 | 0.042 |

Adjustment variable. Age, sex, race, blood alkaline phosphatase, blood urea nitrogen, blood calcium, blood triglycerides, blood phosphorus, blood creatinine, blood uric acid, total active vitamin D, 2,5-hydroxyvitamin D2, presence of hypertension, diabetes mellitus, BMI, and calculated GFR

## Author Contributions

**Conceptualization:** Yuying Jiang.

**Data curation:** Yuying Jiang.

**Formal analysis:** Yuying Jiang.

**Investigation:** Yuying Jiang.

**Methodology:** Yuying Jiang, Xiaorong Bao.

**Project administration:** Xiaorong Bao.

**Software:** Yuying Jiang.

**Validation:** Xiaorong Bao.

**Visualization:** Yuying Jiang.

**Writing – original draft:** Yuying Jiang.

**Writing – review & editing:** Xiaorong Bao.

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
