## [Decision Letter · Decision Letter 0]

6 Mar 2024

PONE-D-24-02266Systemic Immune-inflammatory Indicators and Bone Mineral Density in Chronic Kidney Disease Patients: A Cross-sectional Research from NHANES 2011 to 2018PLOS ONE

Dear Dr. Bao,

Thank you for submitting your manuscript to PLOS ONE. After careful consideration, we feel that it has merit but does not fully meet PLOS ONE’s publication criteria as it currently stands. Therefore, we invite you to submit a revised version of the manuscript that addresses the points raised during the review process.

We look forward to receiving your revised manuscript.

Kind regards,

Ewa Tomaszewska, DVM Ph.D

Academic Editor

PLOS ONE

Reviewers' comments:

Reviewer's Responses to Questions

**Comments to the Author**

1. Is the manuscript technically sound, and do the data support the conclusions?

Reviewer #1: Yes

Reviewer #2: Yes

2. Has the statistical analysis been performed appropriately and rigorously? 

Reviewer #1: Yes

Reviewer #2: Yes

3. Have the authors made all data underlying the findings in their manuscript fully available?

Reviewer #1: Yes

Reviewer #2: Yes

4. Is the manuscript presented in an intelligible fashion and written in standard English?

Reviewer #1: Yes

Reviewer #2: Yes

5. Review Comments to the Author

Reviewer #1: I reviewed the article sent to me for review - "Systemic markers of immune inflammation and bone mineral density in patients with chronic kidney disease: a cross-sectional study from NHANES 2011–2018"

The presented research work may not be groundbreaking in its assumptions, as it has been known for years that CKD and high levels of inflammatory parameters reduce bone mineral density, but the work meets the publication requirements.

I believe that the work is valuable due to the research and breadth of the group analyzed.

Reviewer #2: 1. eGFR needs superscripts for the 2's of the unit species.

2. line 58-59 '[value -0.008; 95% confidence

interval (CI) -0.014, -0.002]' Please change to [β=-0.008; 95% confidence

value -0.008; 95% confidence interval (CI) -0.014, -0.002]'.

3. From table 1 kind of it can be seen that there are very few participants with BMI<18, I suggest that the authors can combine them in the group with BMI<25 only, in addition, the square meters in the units of BMI need to be superscripted.

4. on what basis did the authors choose these covariates?

5. the authors excluded non-compliant participants from the initial group of about 20,000 participants and ended up with about 2,000 participants left to be included in the analysis. does this exclusion result in a selective bias in the sample?

6. It is recommended that the authors add a discussion of SII-related studies to the discussion:

I suggest citing the following literature:

1. doi: 10.3389/fimmu.2022.925690

2. doi: 10.1016/j.numecd.2023.04.015

6. PLOS authors have the option to publish the peer review history of their article (what does this mean?). If published, this will include your full peer review and any attached files.

Reviewer #1: **Yes: **Jakub Kosiński

Reviewer #2: No

---

## [Author Response · Author response to Decision Letter 0]

19 Mar 2024

Dear Editor and Reviewers, 

We appreciate the opportunity to allow us to revise our manuscript and thanks for the reviewers’ constructive comments and suggestions. We would like to submit our revised manuscript, entitled ‘Systemic Immune-inflammatory Indicators and Bone Mineral Density in Chronic Kidney Disease Patients: A Cross-sectional Research from NHANES 2011 to 2018’ for consideration for publication. In the revised manuscript, we have carefully addressed all comments and questions raised by reviewers point-by-point. We greatly appreciate your time and efforts to improve our manuscript for publication.

Reply to Reviewers 

Reviewer #1: 

1． I reviewed the article sent to me for review - "Systemic markers of immune inflammation and bone mineral density in patients with chronic kidney disease: a cross-sectional study from NHANES 2011–2018" The presented research work may not be groundbreaking in its assumptions, as it has been known for years that CKD and high levels of inflammatory parameters reduce bone mineral density, but the work meets the publication requirements. I believe that the work is valuable due to the research and breadth of the group analyzed.

Reply: We sincerely appreciate your acknowledgment of the importance of our study and the paper. We selected SII, a novel parameter combining platelets, neutrophils, and lymphocytes, to quantitatively explore the clinical significance of the inflammatory response in the development of complications in CKD patients and to increase the confidence of the results by expanding the sample size, even though it is already widely known that the combined systemic inflammatory response in CKD patients reduces BMD. We will also investigate the precise mechanisms linking the two in more detail in later research to provide ground-breaking outcomes. Once again, thank you for your confirmation. Please don't hesitate to contact us if you have any further queries or require any clarification.

Reviewer #2: 

Concerns:

1． eGFR needs superscripts for the 2's of the unit species.

Reply: We appreciate your insightful feedback, and we have updated the entire text to include all non-standard units of measurement. Going forward, we'll make sure to focus more on the pertinent aspects of our research. 

2． line 58-59 '[value -0.008; 95% confidence interval (CI) -0.014, -0.002]' Please change to [β=-0.008; 95% confidence value -0.008; 95% confidence interval (CI) -0.014, -0.002]'.

Reply: Thank you very much for your valuable comments, we have revised this part according to the review comments. 

3． From Table 1 kind of it can be seen that there are very few participants with BMI<18, I suggest that the authors combine them in the group with BMI<25 only, in addition, the square meters in the units of BMI need to be superscripted.

Reply: We appreciate your insightful feedback. We have superscripted the square meter needs in terms of BMI and merged participants with a BMI <18 kg/m2 in the BMI <25kg/m2 category. In our next job, we'll also take extra care with these aspects.

. 

4． On what basis did the authors choose these covariates?

Reply: Thank you very much for pointing out this deficiency. During the inclusion of covariates, we considered three aspects that affect the duration of CKD, the body's inflammatory response, and changes in bone mineral density, so we reviewed the relevant literature and referred to our own experience in clinical practice to determine the covariates to be included in the study. We included gender, age, and race as demographic data because this affects the general condition of the patients. Urine albumin, urine creatinine, urine creatinine/albumin ratio, blood urea nitrogen, blood creatinine, blood uric acid, and GFR are often used to assess disease progression in patients with CKD. The presence of hypertension, diabetes mellitus, and obesity, on the other hand, have been shown to influence further deterioration of renal function. Blood alkaline phosphatase, blood calcium, blood triglycerides, blood phosphorus, and vitamin D are closely related to bone metabolism. Taking all these considerations into account, we finally included these covariates.

5． the authors excluded non-compliant participants from the initial group of about 20,000 participants and ended up with about 2,000 participants left to be included in the analysis. does this exclusion result in a selective bias in the sample?

Reply: We much appreciate your candid counsel. We did take into consideration the possibility that, had missing data and patients not diagnosed with CKD been removed from the study because non-randomization was used, there would have been differences between the study population and the target population represented, which would have diminished the validity of the study's conclusions. We increased the sample size by including 8 years of data, spanning from 2011 to 2018, to reduce this mistake. The website also suggests combining multiple years of data to increase the sample size and thereby reduce error, given the history of the NHANES database with the oversampling of certain subgroups and the limited analytical utility of a 2-year sample to provide estimates for subgroups with lower percentages of population distribution. To minimize mistakes, we took this action and included in the study all eligible patients. To lessen confounding in the study and lower the error in the conclusions, we also performed stratified analyses and built multivariate analytic models. We do think about utilizing more public datasets or conducting follow-up research using actual studies to better establish the study's legitimacy. Once again, I appreciate your thoughtful suggestions.

6． It is recommended that the authors add a discussion of SII-related studies to the discussion. I suggest citing the following literature: 

doi: 10.3389/fimmu.2022.925690\\ doi: 10.1016/j.numecd.2023.04.015

Reply: We appreciate you pointing us in the direction of the well-researched literature, which goes into additional detail about the clinical importance of SII in various contexts. Based on your suggestion and after carefully reviewing the research, we have updated the discussion section to provide a fresh analysis of the significance of SII in the diagnosis and prognosis of other diseases. The citations for the sources you suggested are found in lines 342–359. Furthermore, it has been demonstrated that SII is related to abdominal aortic atherosclerosis, a typical consequence of advanced CKD, which offers suggestions for further research. Once again, I appreciate your advice and manuscript recommendation. 

Line 342–357, fourth paragraph of the Discussion section:

According to a 2014 study by Hu [15] et al., SII is a thorough and unique inflammatory biomarker. Prior research has routinely employed SII as a predictor of the development of kidney transplant rejection as well as the incidence of acute or chronic renal injury [34–38]. Numerous studies connected to SII have been conducted in recent years, and they have thoroughly examined its clinical importance. Lai et al. [34] in a cohort study showed that elevated levels of SII before CAG were an important and independent risk factor for postoperative AKI, and Halpern et al. [40] in a single-center cohort study showed that elevated levels of SII were independently associated with increased survival only in post-transplant patients. For instance, Xie et al. [39] in 2022 found that elevated levels of SII were associated with hepatic steatosis but not with hepatic fibrosis. A cohort study was conducted by Halpern et al. in one center. Regarding prognosis, prospective cohort research by Shi [31] et al. showed that in CKD patients with ACS, higher SII was linked to poor cardiovascular outcomes. In a similar vein, Xie et al. [41] discovered a favorable correlation between SII and abdominal aortic calcification, a frequent CKD consequence. In a cohort investigation of critically ill AKI patients, Lan et al. [42] discovered a J-shaped relationship between SII and all-cause death in these patients. 

All of these suggestions are insightful and help us polish our manuscript considerably. We made every effort to make the manuscript better. The paper's structure and content won't be impacted by these modifications. We sincerely thank the editors and reviewers for their hard work, and we hope that the corrections will be accepted. Kindly do not hesitate to get in touch with us if you have any more inquiries or suggestions. 

I express my gratitude once more for your insightful remarks and recommendations.

---

## [Decision Letter · Decision Letter 1]

27 Mar 2024

Systemic Immune-inflammatory Indicators and Bone Mineral Density in Chronic Kidney Disease Patients: A Cross-sectional Research from NHANES 2011 to 2018

PONE-D-24-02266R1

Dear Dr. Xiaorong Bao,

We’re pleased to inform you that your manuscript has been judged scientifically suitable for publication and will be formally accepted for publication once it meets all outstanding technical requirements.

Kind regards,

Ewa Tomaszewska, DVM Ph.D

Academic Editor

PLOS ONE

Additional Editor Comments (optional):

Reviewers' comments:

Reviewer's Responses to Questions

**Comments to the Author**

1. If the authors have adequately addressed your comments raised in a previous round of review and you feel that this manuscript is now acceptable for publication, you may indicate that here to bypass the “Comments to the Author” section, enter your conflict of interest statement in the “Confidential to Editor” section, and submit your "Accept" recommendation.

Reviewer #2: All comments have been addressed

2. Is the manuscript technically sound, and do the data support the conclusions?

Reviewer #2: Yes

3. Has the statistical analysis been performed appropriately and rigorously? 

Reviewer #2: Yes

4. Have the authors made all data underlying the findings in their manuscript fully available?

Reviewer #2: Yes

5. Is the manuscript presented in an intelligible fashion and written in standard English?

Reviewer #2: Yes

6. Review Comments to the Author

Reviewer #2: The authors completed a good scientific research.

I have no further comments.

Thank you so much for your efforts.

7. PLOS authors have the option to publish the peer review history of their article (what does this mean?). If published, this will include your full peer review and any attached files.

Reviewer #2: No

---

## [Editor Report · Acceptance letter]

3 Apr 2024

PONE-D-24-02266R1 

PLOS ONE

Dear Dr. Bao, 

I'm pleased to inform you that your manuscript has been deemed suitable for publication in PLOS ONE. Congratulations! Your manuscript is now being handed over to our production team.

Kind regards, 

on behalf of

Professor Ewa Tomaszewska 

Academic Editor

PLOS ONE